# Bacteriophage and the Innate Immune System: Access and Signaling

**DOI:** 10.3390/microorganisms7120625

**Published:** 2019-11-28

**Authors:** Amanda Carroll-Portillo, Henry C. Lin

**Affiliations:** 1Division of Gastroenterology and Hepatology, University of New Mexico, Albuquerque, NM 87131, USA; carrolla15@gmail.com; 2Medicine Service, New Mexico VA Health Care System, Albuquerque, NM 87108, USA

**Keywords:** bacteriophage, innate immunity, transcytosis, TLR, nucleic acid receptors

## Abstract

Bacteriophage and the bacteria they infect are the dominant members of the gastrointestinal microbiome. While bacteria are known to be central to maintenance of the structure, function, and health of the microbiome, it has only recently been recognized that phage too might serve a critical function. Along these lines, bacteria are not the only cells that are influenced by bacteriophage, and there is growing evidence of bacteriophage effects on epithelial, endothelial, and immune cells. The innate immune system is essential to protecting the Eukaryotic host from invading microorganisms, and bacteriophage have been demonstrated to interact with innate immune cells regularly. Here, we conduct a systematic review of the varying mechanisms allowing bacteriophage to access and interact with cells of the innate immune system and propose the potential importance of these interactions.

## 1. Introduction

The gastrointestinal (GI) microbiome consists of a mix of bacteria, bacteriophage, fungi, and animal viruses. The role of bacteria in structure and function of the microbiome has been the focus of intense research, but until recently, the role of the bacteriophage has not received similar attention. Bacteriophage, or phage, are bacterial viruses. Unlike Eukaryotic and Prokaryotic cells, there is no consensus sequence that allows for easy identification of the bacteriophage population using a molecular approach as compared with 18S and 16S, respectively. Instead, bacteriophage is classified based on morphology, nucleic acid characteristics, and properties [1]. For example, phage can have double-stranded DNA, such as those within the order *Caudovirales*, single-stranded DNA (families *Inoviridae* and *Microviridae*), or RNA (families *Leviviridae* and *Cystoviridae*) [2,3]. Bacteriophage undergoes two types of life cycles, lytic or lysogenic [4]. The lytic life cycle involves adsorption of a phage onto its bacterial host, infection of the host with the viral nucleic acid, hijacking the replication machinery to generate new virions, and then lysis of the host bacteria for release of phages into the surrounding environment. The lysogenic life cycle begins similarly, but rather than propagating new virions, the phage nucleic acid becomes incorporated into the bacterial genome to exist as a latent prophage. It remains in this state until certain triggers such as DNA damage, reactive oxygen species, temperature changes, etc., cause the phage to excise itself from the bacterial genome where it can then function extra-chromosomally [5] or propagate progeny using the host machinery, and lyse the host for extracellular release [6,7,8]. While the lytic cycle allows for rapid increase in phage numbers with concurrent decrease in bacterial numbers, lysogeny allows for phage manipulation of bacterial gene expression without a loss of the bacterial host [9,10,11,12,13,14].

In nature, phages are found wherever bacterial communities thrive. Most of the information we have on phage function comes from research of marine phages in their natural environment [15]. In the ocean, phages outnumber marine bacteria by approximately 10-fold [16], and the majority are lytic [17]. Within the GI tract, lysogeny is the predominant phage lifestyle [18,19] with bacteria within the phyla Proteobacteria and Firmicutes harboring more prophages than those of Bacteroidetes or Actinobacteria [20,21]. Intrapersonal variation of the GI bacteriophage population is low with less variation occurring over time, but interpersonal variation between individuals is high, even when intestinal bacterial communities are similar [9,18,19,22,23]. In animals, bacteriophage abundance in mucosal environments is estimated to be about 4-fold higher than that of bacteria [24].

Although the main abundance of bacteriophage is found within the GI microbiome, phages are also capable of moving outside of the GI tract [25,26]. The large numbers of phages and their ubiquitous presence within the body allows for ample opportunity for the Eukaryotic host to sense their presence and initiate an immune response. Initial contact and response to phage by the Eukaryotic host occurs through the cells of the innate immune system. The innate immune system is comprised of phagocytes (macrophages and dendritic cells (DCs)), granulocytes (eosinophils, basophils, neutrophils, natural killer cells, and mast cells), as well as a complement protein, which allows for opsonization of foreign particles permitting increased phagocytic uptake [27]. The cells of the innate immune system share the ability to sense microbe-specific material, such as bacterial endotoxin and that of bacteriophage, through a family of receptors called pathogen recognition receptors (PRR) that recognize pathogen-associated molecular patterns (PAMPs). Included in this family are the Toll-like receptors (TLRs) and nucleic acid receptors (CDN sensors, AIM2-like receptors, and RIG-1-like receptors). TLRs recognize pathogen epitopes extracellularly (TLR1, 2, 4, 5, 6, and 11) and endosomally (TLR3, 7, 8, and 9 recognize nucleic acid) [28,29]. Cytoplasmic receptors that signal in response to cytoplasmic DNA and RNA include Retinoic acid-inducible gene-I (RIG-1)-like receptors, absent in melanoma 2 (AIM2)-like receptors, and cyclic-di-nucleotide (CDN) sensors [30,31,32]. Upon recognition of their cognate ligand, PRRs signal to stimulate either a pro-inflammatory or an anti-inflammatory response.

In the case of most bacterial infections and in the setting of an altered gastrointestinal microbiome (dysbiosis), inflammation is responsible for tissue damage and development of disease phenotypes. Interaction of bacteria and their PAMPs with innate immune cells are well studied [33,34], but the interactions between bacteriophage and these same cells are less understood. A number of Eukaryotic cells express PRRs [35,36,37] allowing for the possibility of bacteriophage recognition at multiple locales within the host. Among them, cells of the innate immune system (i.e., dendritic cells and macrophages) are likely central to recognition of mucosal-associated bacteriophage. Here, we review the manner in which bacteriophage may access innate immune cells and the likely mechanisms for recognition of bacteriophage PAMPs by these same cells.

## 2. Accessing Innate Immune Cells

Bacteriophages move beyond the mucosal environments they inhabit, and have been found in the bloodstream, spleen, kidney, liver, and brain [9,25,38]. As bacteriophages are microorganisms, they possess a number of epitopes that would be recognized by the PRR of the innate immune system and have been shown to stimulate anti-phage antibody production [39,40,41]. In order to initiate immune responses driven by innate immune cells such as dendritic cells and macrophages, bacteriophages must first come into contact with them.

Innate immune cells, including macrophages, dendritic cells, and mast cells that are part of the gastrointestinal mucosal immune system are localized to the gastrointestinal mucosa. These represent the first immune cells that bacteriophages would encounter. Phages are more abundant at mucosal surfaces with some phage expressing protein domains specific for mucin sugar residues [24]. Coliphagic T4 binds oligosaccharide side chains of mucin proteins through small domains that are part of the capsid proteins allowing a fixed locale to increase likelihood of bacterial encounters. Other *Caudovirales* phages have similar binding domains within their capsid proteins. This form of anchoring within the mucin likely increases opportunity for phage–bacterial contact, and in turn, the opportunity to infect or lysogenize their host.

There are a number of ways by which phages may cross the gastrointestinal epithelium (transcytosis [42]) (Figure 1) including: (1) Free uptake, where phage alone is endocytosed and transported across [43,44,45,46]; (2) trojan horse, where a phage with its infected bacterial host enters or is endocytosed together [45,47]; or (3) crossing via a leaky gut, where inflammation or injury has resulted in impaired barrier function allowing for passive transit of phage through the epithelium [25,48,49].

To date, Nguyen et al. [50] has performed the most comprehensive investigation of free uptake of bacteriophages by eukaryotic cells. In their work, they examined the ability of *Caudovirales* phage (from the families *Myoviridae*, *Siphoviridae*, *Podoviridae*) specific to different bacterial hosts to translocate across epithelial or endothelial cell lines. They found that phage transport occurred preferentially in an apical to basolateral direction and at maximum, 0.1% of applied phages moved across any monolayer. During transcytosis, phages could be found within endosomal compartments. Disruption of the Golgi apparatus, and only the Golgi, inhibited transit. Of interest, inhibition of receptor-mediated endocytosis with Wortmannin treatment or inhibition of clathrin-dependent endocytosis with chloroquine treatment did not inhibit phage transit, suggesting that endocytosis of free T4 through the epithelial barrier may be passive. In this study, paracellular transcytosis of T4 was not observed. This result contrasted with an observation of an experiment with M13 phage where Ivanenkov et al. [51] reported that transport of M13 phage was blocked by chloroquine treatment suggesting the involvement of a clathrin-dependent endocytic pathway for transcytosis of this phage. Within the group of phages tested by Nyugen et al. [50], there was substantial variation of the percentage of each type of phage that successfully transcytosed an MDCK (canine kidney epithelial) monolayer. In addition, T4 phage transcytosis varied depending on the type of cell monolayer tested. These data indicated that transcytosis is likely both phage- and tissue-specific so that certain phages are able to move across specified tissues more easily while others, not at all. Other data suggesting a specificity of phage to epithelial cells come from Shan et al. [52] who examined the ability of three different *Clostridium difficile* phages to adhere to either HT29 (human colonic) or HeLa (human cervical) monolayers. They found that two phages adhered to HT29, one better than the other, while the third did not adhere at all. None of the phages tested adhered to HeLa cells.

Another mechanism of phage transcytosis involves transport of phage across the barrier while residing within a bacterium (Trojan horse). Phages can be shuttled across the mucosal barrier within the bacterium using the usual pathways of bacterial transcytosis [53,54]. Some phages actually increase the likelihood of transcytosis by increasing bacterial aggregation at epithelial surfaces. Bille et al. [55] showed that presence of a filamentous phage resulted in increased aggregation of *Neisseria meningitidis* at epithelial surfaces by forming a linker between the cells and bacteria. Phage lysogeny may result in upregulation of virulence factors [56,57] including secretion systems, biofilm genes, and toxins that could all potentially increase bacterial uptake through the epithelial layer.

Transport through mucosal epithelial cells or M cells in the GI tract is not the only means by which phages may access the underlying immune cells. The immune cells themselves are capable of sampling the luminal environment as dendritic cells have been shown to reach through tight junctions of the epithelial barrier in order to phagocytose luminal material [58,59]. Both free phages, conveniently embedded within the mucus, and a phage within an infected bacterium can be taken up for transport through the mucosal surface in this manner. Phage transcytosis within dendritic cells might also serve as a mechanism of down regulating subsequent immune response to the phage. Barfoot et al. [60] demonstrated that while dendritic cells rapidly phagocytose T4 phage, this process results in inhibition of any subsequent phagocytosis suggesting an overall decrease in phage uptake.

The final manner in which phages might transcytose the mucosal epithelium occurs in the setting of dysbiosis-induced leaky gut. In instances of leaky gut, openings occur between cells where tight junctions previously prevented paracellular movement of organisms (bacterial or viral) across the mucosal barrier. In a healthy mucosal barrier, paracellular transport is limited to molecules that pass via either the pore or leak pathways [61]. These pathways are limited to 5–10 Angstroms and below ~62.5 Angstroms, respectively; both well below the size of bacteriophage. Inflammation, injury, or microbial signals could cause increased permeability of the epithelial barrier by influencing the tight junctions. In this scenario, access to the underlying immune cells could now also be granted to phages that might not transcytose under healthy conditions. These phages, in turn, might stimulate an increase in damaging pro-inflammatory responses and contribute to further development of disease states. Tetz et al. [62] demonstrated that healthy rats treated only with mixed bacteriophage solutions developed symptoms of leaky gut with concurrent shifts in the gut microbial populations consistent with those seen in clinical dysbiosis. The authors suggest bacteriophage effect on mucosal barrier integrity occurs indirectly through modulation of the bacterial constituents of the microbiome. However, their data do not rule out a potential direct effect by bacteriophage on the epithelial barrier.

Our understanding of phage biology has been limited by how we have designed our experiments. For example, phage transcytosis has primarily been studied with a single type of bacteriophage when there are numerous types of bacteriophage within the microbiome and diversity could play a role in the interaction occurring at the mucosal surface. Further investigation into which phages transcytose, and under what circumstances, is necessary in order to better understand phage’s contribution to mucosal immunity.

## 3. Signaling Innate Immune Cells

As bacteriophage are not known to infect mammalian cells, one might ask why Eukaryotic cells have mechanisms for sensing these bacterial viruses. The body of literature investigating the interactions of bacteriophage with innate immune cells suggests that these interactions represent bacteriophage signaling through the same pathways used for sensing animal viruses, and in most cases, doing so to affect a dampening of the inflammatory response that has been stimulated by the presence of bacteria [63,64]. There are instances of phage signaling being beneficial to the Eukaryotic host in the event of bacterial infection [65] as well as instances of phage signaling being beneficial to the bacterial pathogen [63], indicating that results of phage signaling are very specific to the instances in which they occur. The physiological effects of bacterial signaling through PRRs warrants further examination, but here we are focusing solely on the mechanisms in which this immune signaling may occur.

Immune signaling first requires activation through ligand binding with extracellular or intracellular receptors. Once bacteriophages have gained access to the cells of the mucosal innate immune system, there are a number of mechanisms through which they could be recognized (Figure 2). These mechanisms can be broken down into three main categories: (1) Extracellular recognition, (2) endocytic recognition, and (3) cytoplasmic recognition. Extracellular recognition of a free phage is likely to occur either through binding of cell surface moieties, integrins, or currently unknown phage receptors [66,67]. *C. difficile* phage adhered to epithelial cells in a cell-specific, phage-specific manner [52], and Gorski et al. [66] identified Lys-Gly-Asp (KGD) domains in the T4 capsid proteins with homologs involved in integrin binding [68,69] suggesting possible phage–integrin interactions. In addition to KGD domains, T4 and other members of *Caudovirales* also possess capsid proteins with Ig-like domains [70]. The Hoc protein of the T4 capsid contains three of these domains with homology to either I-set or PKD-like domains [71]. These domains are known to effectuate mucin binding by phage [24]. It is likely that both the KGD motif and Ig-like domains function for phage adhesion to cells outside the mucosal environment. Eukaryotic homologs of the KGD motif participate in integrin interactions [68] and the Ig-like domains are known to function in adhesive protein-protein or protein-ligand interactions [72,73].

Endocytic recognition begins with endocytosis of either free phage, opsonized phage (with bound antibody and complement), or infected bacteria followed by movement along the lysosomal pathway [43,60,74,75]. Degradation of the lysosomal contents mean that phage epitopes are exposed to PRR within the endosomal compartment. In the case of lysogenized bacteria, this requires degradation of the bacterial membranes. The incorporated phage DNA is then released along with the bacterial DNA. For phage infected bacteria and free bacteriophage, degradation of the viral capsid/proteins releases the phage nucleotides allowing for PRR sensing. The PRR include a number of endosomally associated TLRs that sense nucleic acids, including TLR3 (binds dsRNA and poly I:C), TLR7 and 8 (bind ssRNA and poly dT), and TLR9 (binds CpG DNA) [76,77,78]. Gogokhia et al. [79] showed that coliphage isolated from human fecal material stimulated mouse dendritic cells through the TLR9 pathway. This effect extended to two other DNA phage (*L. plantarum* and *B. thetaiotamicron* phage) as well. Sweere et al. [80] demonstrated that the *Pseudomonas* filamentous phage Pf, an RNA phage, signals through the TLR3 pathway upon endocytosis by mouse bone marrow derived dendritic cells. Application of bacteriophage derived dsRNA to human peripheral blood mononuclear cells also stimulated immune signaling through TLR3 activation, but resulted in downstream signaling that was different from signaling initiated by Poly I:C activation of TLR3 [81], demonstrating that ligands for the same receptor can still effectuate different immune signaling. While there is currently no experimental evidence of phage activation of TLR7/8, ssRNA phage belong to the family *Leviviridae* and are responsible for infecting enterobacteria found within the microbiome. Infection of enterobacteria increases the likelihood that *Leviviridae* phage also interact with innate immune cells and likely do so through TLR7/8.

Cytoplasmic recognition of phage DNA within a eukaryotic host has not yet been described but is most likely to occur with a phage that infects intracellular bacteria such as *Chlamydia*, *Mycobacterium*, and *Listeria*. The most complicated part of this activation pathway is access of the phage nucleic acid to the cytoplasmic space. The epitopes of intracellular bacteriophage would likely become exposed in the cytoplasm upon either bacterial death or phage-induced bacterial lysis. An example of this possibility was demonstrated by Hsia et al. [47] where infection of *Chlamydia psittaci* with phage ΦCPG1 resulted in burst of an intracellular, phage-containing endosomal compartment with subsequent release of phage particles into the cytoplasm. Phage DNA in the cytoplasm could be recognized by nucleotide sensors such as cGAS-STING or the AIM2-like receptors [82,83]. Phage RNA could signal through RIG-1-like receptors (RIG-1 or MDA5) [76,77,78,79,80]. Recently, Cohen et al. [84] reported a bacterial homolog to the cGAS-STING pathway within bacteria that is specific for phage sensing. This bacterial homolog is a likely a precursor to the eukaryotic system that still might retain phage-sensing abilities. RIG-1-like receptors are reported to require active replication in the cytoplasm [85] and have thus been largely ruled out for sensing of phage DNA as phages are not believed to replicate within the eukaryotic host. While there is currently no direct evidence of phage replication occurring within a Eukaryotic host, there are tempting indicators in the literature suggesting it may occur. For example, Eukaryotes possess capsid gene orthologs in their genome that resemble those encoded by *Chlamydia pneumoniae* phage [86]. Eukaryotes express homologs to bacterial aerolysin and lysozyme suggesting gene transfer could have occurred, an ability commonly associated with bacteriophage [87,88]. Finally, phages of *Wolbachia* (an obligate intracellular bacterium) contain metazoan genes [89]. These examples would require active phage processes within the Eukaryotic host allowing for exchange of genetic material.

## 4. Summary and Future Directions

Bacteriophage function in regulation of bacterial populations are well studied, but within the mucosal environments that bacteriophage inhabit, they play a larger role influencing the function of the eukaryotic host through activation of the immune system [79,90,91]. Cells of the innate immune system reside within the mucosal tissues and are essential for sensing self and non-self to initiate regulatory signaling pathways. Outside dendritic cell luminal sampling, phages must first transcytose across the mucosal barrier to access the immune cells with which they interact. Transcytosis occurs through a variety of mechanisms that are likely to be different depending on the phage. Proximity of mucosal macrophages, dendritic cells, and mast cells means immune signaling is accessible for phage immediately after transcytosis. Additionally, phage circulation throughout the body means there are immune cell interactions possible at distant sites including with Kupffer cells, PBMC, and splenocytes. Upon contact, bacteriophage can stimulate immune signaling through extracellular, endosomal, or cytoplasmic PRR that recognize specific microbial epitopes like those expressed by phage.

Would there be a purpose of bacteriophage interacting with the metazoan immune system if these viruses only infect bacteria? This question gets at the heart of bacteriophage research. Although the idea that phages are simply bacterial viruses, each with a very narrow host range, is a long-standing paradigm, we have actively selected and propagated phages to act against a single type of bacteria. Could that approach have masked the abilities of bacteriophage? Could the limited host range of bacteriophage be a function of the techniques used in their isolation, as suggested by Ross et al. [92]? In addition, experiments on the effects of bacteriophage often fail to account for population dynamics, a factor that is likely to be a major contributor to overall phage biology. Bacteriophages do not exist in the environment as a monolithic group of a single discrete type. Instead, the natural environment supports not only a diverse bacteriophage population within their niche, but a diverse bacterial community as well. Imagine the modulating effect of selective pressures on bacteriophage biology. Could our understanding of phage biology be correct when most experiments investigated an artificial group consisting of only a single type of phage? Consistent with the idea that our experimental approaches may have masked the true nature of phage biology, is the finding by a number of groups that a bacteriophage population is capable of broadening their host range [93,94,95]. Such findings would argue away from the current paradigm that bacteriophage targets only a narrow, fixed range of bacteria.

Could the existence of interaction between bacteriophage and the metazoan immune system be similarly masked by our experimental design? While a limited expectation would propose that the immune system might develop an immune understanding of the native bacteriophage population at mucosal surfaces, could bacteriophage interact with cells in a manner more reminiscent of animal viruses? Specifically, could phages better their own survival by repressing immune signaling that is detrimental to their bacterial host? More research is needed to explore the possibility of phage–metazoan host cell interactions.

In addition to better understanding of the outcomes of phage–metazoan interactions, there are many directions that research specifically into bacteriophage interaction with the innate immune system can move. Of particular interest are questions surrounding bacteriophage transcytosis, including determination of which bacteriophages transcytose and which pathway they pursue. Another line of query includes the mechanisms free phages use not only for transcytosis, but for phagocytosis by innate immune cells. Do these mechanisms vary from phage to phage and does that variance ultimately influence how these phages are sensed and what signaling pathway they stimulate? There are multiple opportunities to better characterize the role a diverse phage population plays in maintenance of gastrointestinal homeostasis. Just imagine.

## Figures and Tables

**Figure 1 microorganisms-07-00625-f001:**
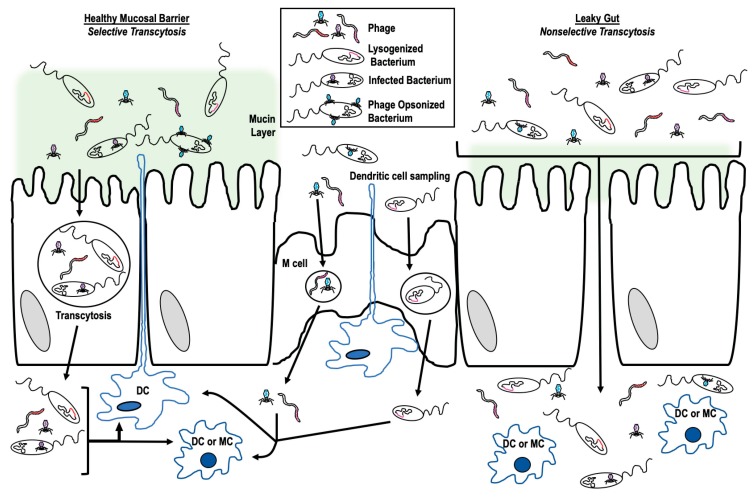
Mechanisms of bacteriophage transcytosis to access innate immune cells. Graphical representation of the variety of ways that bacteriophage may use to cross the mucosal epithelial barrier. Bacteriophage nucleic acid is denoted in color to differentiate kinds of phage and indicate instances of bacterial infection or lysogeny. Dendritic cells (DC) and macrophages (MC) are labeled.

**Figure 2 microorganisms-07-00625-f002:**
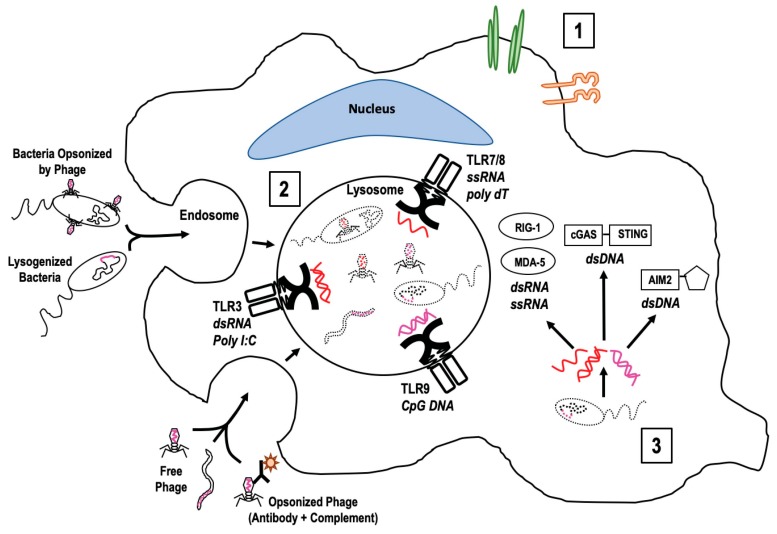
Potential mechanisms of bacteriophage recognition by innate immune cells. Pictorial representation of three ways that bacteriophage may be recognized by cells of the innate immune system to initiate immune signaling. 1. Extracellular recognition by integrins, membrane moieties, or phage-specific receptors, 2. Endosomal recognition by endosomal PRRs after phage in any form is endocytosed and degraded within the lysosomal compartment, and 3. Cytoplasmic recognition of bacteriophage nucleotides through cGAS-STING, RIG-1-like receptors (RIG-1, MDA5), or AIM2-like receptors. Phage nucleic acid is colored (red = DNA, pink = RNA).

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
