# Peer review of "Bacteriophage and the Innate Immune System: Access and Signaling"

_microorganisms, 2019, doi:10.3390/microorganisms7120625_

Round 1

Reviewer 1 Report

The interaction among  innate immune system and bacteriophage is a very interesting topic, and due to the enormous number of bacteriophages and the relative low amount of information that have been collected up to now, probably we are only looking at the tip of an iceberg. This review summarizes the recent literature on this topic, underlying that there are several open questions and unknown actors in these interactions.

Among the mechanisms used to cross the mucosal epithelial barrier the authors mention the paracellular transcytosis, but they do not describe it. It would increase the understanding of the text if the authors can spend a few words of explanation about this kind of transcytosis

There are minor texts inconsistency:

Pag 1 Lane 16: the last sentence seems lacking one or more words

Figure 1 legend: Graphical representation of the variety of ways that bacteriophage may cross the mucosal….should be Graphical representation of the variety of ways that bacteriophage may use to cross the mucosal

Author Response

 We thank the reviewer and Editor for their supportive comments and insightful suggestions that have helped to make the revised manuscript stronger.

The interaction among innate immune system and bacteriophage is a very interesting topic, and due to the enormous number of bacteriophages and the relative low amount of information that have been collected up to now, probably we are only looking at the tip of an iceberg. This review summarizes the recent literature on this topic, underlying that there are several open questions and unknown actors in these interactions.

Among the mechanisms used to cross the mucosal epithelial barrier the authors mention the paracellular transcytosis, but they do not describe it. It would increase the understanding of the text if the authors can spend a few words of explanation about this kind of transcytosis

Corrected.  It is more likely that phages cross the epithelial layer via a paracellular transport route only in the abnormal setting of a leaky gut.  As such, we have changed our figure so that paracellular transport is now found only on the “Leaky Gut” side of Figure 1 where modification of the tight junctions has occurred allowing for transport of larger material in a non-regulated fashion. This is consistent with Tetz, G.V. et al. (2017, 7, Sci Reports) who demonstrated that treatment of rats with bacteriophage cocktails resulted in increased barrier leakage and Roxas, J.L., et. al. (2018, 8, Comprehen Physiol) who reported that the size limitations of paracellular transport across a healthy mucosal barrier would likely rule out transport of bacteriophage.

There are minor texts inconsistency:

Pag 1 Lane 16: the last sentence seems lacking one or more words

Corrected. We have added words to clarify the sentence

Figure 1 legend: Graphical representation of the variety of ways that bacteriophage may cross the mucosal….should be Graphical representation of the variety of ways that bacteriophage may use to cross the mucosa

 Corrected.

Reviewer 2 Report

In the review by Carroll-Portillo and Lin, the authors provide a brief review of mammalian host responses to bacteriophages. They focus on describing how phages would come in contact with mammalian cells to activate immunity, since phages themselves do not infect mammalian cells like other viruses, and they describe the types of receptors that would activate a specific type of innate immune response. Overall, the review is well-written, inclusive, and easy to read. Below are a few major and minor points to be addressed prior to manuscript acceptance.

Major:

Why would a phage activate a mammalian immune response if the phage itself is not infectious to mammalian cells. Alternatively, why would the mammalian host expend energy and resources on initiating an immune response to microorganisms that are not infectious? For example, the mammalian host is self-tolerant to commensal bacterial, so shouldn’t also be tolerant to phage? Page 4, lines 27-28, poses a very interesting question – if the experiments on phage-host interactions are limited due to experimental design, are there certain conclusions about phage biology that may not be correct? Page 6, lines 1-6: The authors state that bacterial lysis results in phage DNA or RNA to be sensed. Wouldn’t the phage need to be lysed or its genome unpackaged for the mammalian host to sense phage nucleic acids? Page 6, lines 11-12: This statement should be better referenced. At the crux of this review is how/why phages can be sensed by mammalian cells to initiate immunity. If they are able to replicate in mammalian cells, this needs to be described in more detail.

Minor:

Page 2, line 35: Why are only immune cells considered here? PRRs can be found on any type of nucleated cell, e.g., fibroblasts, hepatic cells, kidney cells, epithelial cells. Page 4, line 11: Can phages be transported across mucosal epithelial cells? Typically, microorganisms only gain access to the gut lumen via microfold (M) cells. Figure 2 caption: “Cytoplasmic recognition by CDNs” does not make sense. cGAS sensed cytosolic DNA and then metabolizes cGAMP for detection by STING. Also, why does the figure show mRNA in contact with cGAS-STING. These immune sensors do not sense mRNA. Page 6, line 5: AIM2 does not sense CDNs.

Author Response

 We thank the reviewer and Editor for their supportive comments and insightful suggestions that have helped to make the revised manuscript is stronger.

Major:

Why would a phage activate a mammalian immune response if the phage itself is not infectious to mammalian cells. Alternatively, why would the mammalian host expend energy and resources on initiating an immune response to microorganisms that are not infectious? For example, the mammalian host is self-tolerant to commensal bacterial, so shouldn’t also be tolerant to phage?

The interaction between phage and innate immune cells may be the result of the phage dampening the immune response against bacteria. For example, macrophages treated with both T4 and E. coli have a dampened inflammatory response as compared to E. coli alone. Pretreatment of cells with phage results in decreased uptake of E. coli  (Przerwa et al., 2006, 195(3), Med Microbiol Immunol). Such dampening effect is consistent with tolerance.  We have now included a brief discussion (Page 4, lines 31-41).

Page 4, lines 27-28, poses a very interesting question – if the experiments on phage-host interactions are limited due to experimental design, are there certain conclusions about phage biology that may not be correct?

We believe this is the case. Experiments investigating the effects of mixed populations of phage suggest that there is a broadening of host recognition among the phage population that occurs under certain circumstances. For example, Tetz et al.(2017, 7, Sci Reports) reported that while their bacteriophage mixtures were very specific to certain types of bacteria (i.e. Salmonella), the bacteria affected by phage were not limited to just those strains. Thus, published findings already contradict the current idea that individual phages are only capable of infecting a single targeted type of bacterium.

Page 6, lines 1-6: The authors state that bacterial lysis results in phage DNA or RNA to be sensed. Wouldn’t the phage need to be lysed or its genome unpackaged for the mammalian host to sense phage nucleic acids?

The reviewer is correct, when lysogenized bacteria are degraded in the lysosome, the incorporated bacteriophage DNA is released along with the bacterial DNA allowing for sensing via the TLRs. We have added text to clarify this point (page 5, lines 11-14).

Page 6, lines 11-12: This statement should be better referenced. At the crux of this review is how/why phages can be sensed by mammalian cells to initiate immunity. If they are able to replicate in mammalian cells, this needs to be described in more detail.

Corrected.  We have added a new reference as well as additional text discussing the potential for phage activity within the mammalian host cells (Page 6, lines 19-21 and lines 25-26).

Minor:

Page 2, line 35: Why are only immune cells considered here? PRRs can be found on any type of nucleated cell, e.g., fibroblasts, hepatic cells, kidney cells, epithelial cells.

Agreed.  We have now added text to explain that while the focus of this review is on cells of the innate immune system, PRRs are more broadly on a variety of nucleated cells (Page 2, Lines 27-30 and Page 2, Lines 38 and 39)

Page 4, line 11: Can phages be transported across mucosal epithelial cells? Typically, microorganisms only gain access to the gut lumen via microfold (M) cells.

 While there is not currently in vivo support for this method of transport, we have now cited Nyugen et al. (2017, 8(6), MBio) for their report of transport of phage across epithelial cells. They found bacteriophage transit across a number of different cell monolayers in vitro that occurred preferentially in an apical to basal manner and did not occur via paracellular transit. This is discussed in the text (page 3, lines 5-14).

Figure 2 caption: “Cytoplasmic recognition by CDNs” does not make sense. cGAS sensed cytosolic DNA and then metabolizes cGAMP for detection by STING. Also, why does the figure show mRNA in contact with cGAS-STING. These immune sensors do not sense mRNA. Page 6, line 5: AIM2 does not sense CDNs. 

 Corrected.  We have now made the corrections as suggested by removing mention of CDNs and changing “mRNA” to “dsDNA”. Page 6, Line 13 removed CDN and replaced with “nucleotide”. In line 18, we added reference showing that active transcription is required for sensing.

Round 2

Reviewer 2 Report

The revised manuscript has been greatly improved.  However, one question remains.  Both reviewers raised a similar point regarding experiments involving host-phage interactions and transcytosis of phages across gut epithelium.  Their response to these points discussed research in Tetz et al., 2017 and Roxas et al., 2018; however, neither their response nor the citations are included in the revision.

Author Response

We have added discussion and citing information for both references on page 4, lines 19-32.